# Structural basis for the recognition of LDL-receptor family members by VSV glycoprotein

Jovan Nikolic [1], Laura Belot[1], Hélène Raux[1], Pierre Legrand [2], Yves Gaudin [1] & Aurélie A. Albertini[1]

Vesicular stomatitis virus (VSV) is an oncolytic rhabdovirus and its glycoprotein G is widely used to pseudotype other viruses for gene therapy. Low-density lipoprotein receptor (LDL-R) serves as a major entry receptor for VSV. Here we report two crystal structures of VSV G in complex with two distinct cysteine-rich domains (CR2 and CR3) of LDL-R, showing that their binding sites on G are identical. We identify two basic residues on G, which are essential for its interaction with CR2 and CR3. Mutating these residues abolishes VSV infectivity even though VSV can use alternative receptors, indicating that all VSV receptors are members of the LDL-R family. Collectively, our data suggest that VSV G has specifically evolved to interact with receptor CR domains. These structural insights into the interaction between VSV G and host cell receptors provide a basis for the design of recombinant viruses with an altered tropism.

[1] Institute for Integrative Biology of the Cell (I2BC), CEA, CNRS, Univ. Paris-Sud, Université Paris-Saclay, 91198 Gif-sur-Yvette cedex, France. [2] Synchrotron SOLEIL, 91192 Gif-sur-Yvette cedex, France. These authors contributed equally: Jovan Nikolic, Laura Belot. Correspondence and requests for materials should be addressed to Y.G. (email: yves.gaudin@i2bc.paris-saclay.fr) or to A.Albertini. (email: aurelie.albertini@i2bc.paris-saclay.fr)

Vesicular stomatitis virus (VSV) is an enveloped, negative-strand RNA virus that belongs to the Vesiculovirus genus of the Rhabdovirus family. It is an arbovirus which can infect insects, cattle, horses, and pigs. In mammals, its ability to infect and kill tumor cells, although sparing normal cells makes it a promising oncolytic virus for the treatment of cancer[1–3]. VSV genome encodes five structural proteins among which a single-transmembrane glycoprotein (G). G plays a critical role during the initial steps of virus infection[4]. First, it is responsible for virus attachment to specific receptors. After binding, virions enter the cell by a clathrin-mediated endocytic pathway[5,6]. In the acidic

environment of the endocytic vesicle, G triggers the fusion between the viral and endosomal membranes, which releases the genome in the cytosol for the subsequent steps of infection. Fusion is catalyzed by a low-pH-induced large structural transition from a pre toward a post-fusion conformation, which are both trimeric[7,8].

The polypeptide chain of G ectodomain folds into three distinct domains which are the fusion domain (FD), the pleckstrin homology domain (PHD), and the trimerization domain (TrD). During the structural transition, the FD, the PHD, and the TrD retain their tertiary structure. Nevertheless, they undergo large

**Fig. 1** VSV G interacts specifically with CR2 and CR3 in its pre-fusion conformation. **a** Scheme of the modular organization of the LDL-R indicating the 7 CR modules (1–7), the 3 EGF repeats (a,b and c) , the seven-bladed β-propeller domain (β) of the epidermal growth factor precursor like domain (EGF), and the C-terminal domain containing O-linked oligosaccharides (O-link). SP signal peptide, TM transmembrane domain. **b** SDS–PAGE analysis of interaction experiments between the 7 GST-CR proteins, bound to GSH magnetic beads, and $G_{th}$ at pH 8. **c**, **d** Coomassie-stained SDS–PAGE of interaction experiments between GST-CR1, GST-CR2 and GST-CR3, bound to GSH magnetic beads, and $G_{th}$ (**c**) or VSV (**d**) at pH 8 and 6, respectively. Purified GST-CR bound to GSH magnetic beads were incubated with either $G_{th}$ or VSV in the appropriate pH condition in presence of $Ca^{2+}$ for 20 min at 4 °C. Then, after wash, the beads were directly loaded on a gel. As a control in **b**, GST alone bound to the GSH coated beads was incubated in presence of $G_{th}$. **e** Cartoon that illustrates the experiments presented in **f** and **g**. After 4 h of infection, BSR cells were labeled with an antibody directed against VSV nucleoprotein (anti-VSV N) to visualize the infection (green fluorescence) and a GST-CR$^{ATTO550}$ to probe CR domain recognition by the surface displayed glycoprotein (red fluorescence). **f** Labeling of G at the surface of BSR cells infected with VSV using fluorescent GST-CR1$^{ATTO550}$, GST-CR2$^{ATTO550}$, and GST-CR3$^{ATTO550}$. At 4 h post-infection (p.i.), cells were incubated with the appropriate GST-CR$^{ATTO550}$ at 4 °C during 30 min prior fixation and permeabilization and then immuno-labeled using an anti-VSV N antibody to visualize the infection. **g** Labeling of CHAV G at the surface of BSR cells infected with a recombinant VSV expressing CHAV G (VSVΔG-CHAVG) using fluorescent GST-CR2$^{ATTO550}$ and GST-CR3$^{ATTO 550}$. Infected cells are labeled using anti-VSV N antibodies. In **f** and **g**, DAPI was used to stain the nuclei. Scale bars=20 µm

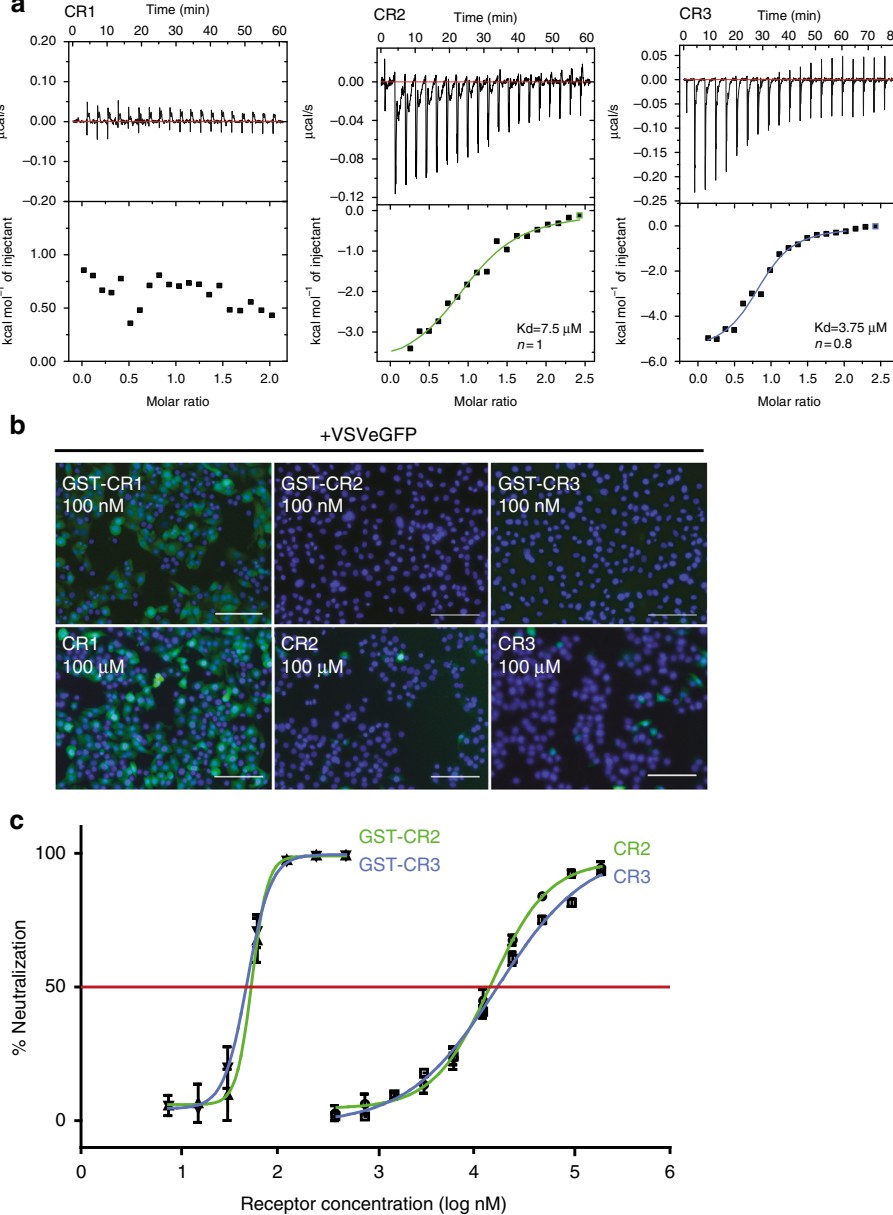

**Fig. 2** Characterization of VSV G-CR2 and VSV G-CR3 interaction **a** Isothermal titration calorimetry (ITC) analysis between $G_{th}$ and CR1, $G_{th}$ and CR2, $G_{th}$ and CR3 at 20 °C. Representative plots of each ITC experiments are illustrated with raw data in the upper panel. Binding parameters were determined by curve fitting analysis with the single-site binding model. The values indicated in the panel are those corresponding to the curves that are presented. $K_d$ values given in the text are means of three independent experiments±standard errors. **b**, **c** Inhibition of VSV infection by soluble forms of CR domains. **b** BSR cells were infected with VSV-eGFP preincubated with GST-CR1, GST-CR2, GST-CR3 (upper part), CR1, CR2, or CR3 monovalent domains (lower part) at the indicated concentrations. Cells were fixed 4 h p.i. Only infected cells are expressing eGFP. Note that neither CR1 nor GST-CR1 construction protect cells from infection. DAPI was used to stain the nuclei. Scale bars=100 μm. **c** VSV-eGFP was preincubated with increasing concentrations of GST-CR2, GST-CR3, CR2, or CR3 monovalent domains. At 4 h p.i., the percentage of infected cells was determined by counting the number of cells expressing eGFP using a flow cytometer. The percentage of neutralization was equal to 100 × [1−(% of infected cells in presence of CR)/(% of infected cells in the absence of CR domains)]. Data depict the mean with standard error for experiments performed in triplicate

rearrangements in their relative orientation due to secondary changes in hinge segments (S1 to S5), which refold during the low-pH induced conformational change[7–10].

VSV G has been widely used for pseudotyping other viruses[11–13] and VSV-G-pseudotyped lentiviruses (VSV-G-LVs) exhibit the same broad tropism as VSV. Recently it has been shown that low-density lipoprotein receptor (LDL-R) and other members of this receptor family serve as VSV receptors[14]. This result explains why VSV-G-LVs do not allow efficient gene transfer into unstimulated

T cells, B cells, and hematopoietic stem cells, as they have a very low expression level of LDL-R[15].

The LDL-R is a type I transmembrane protein which regulates cholesterol homeostasis in mammalian cells[16]. LDL-R removes cholesterol carrying lipoproteins from plasma circulation. Ligands bound extracellularly by LDL-R at neutral pH are internalized and then released in the acidic environment of the endosomes leading to their subsequent lysosomal degradation. The receptor then recycles back to the cell surface. LDL-R ectodomain is

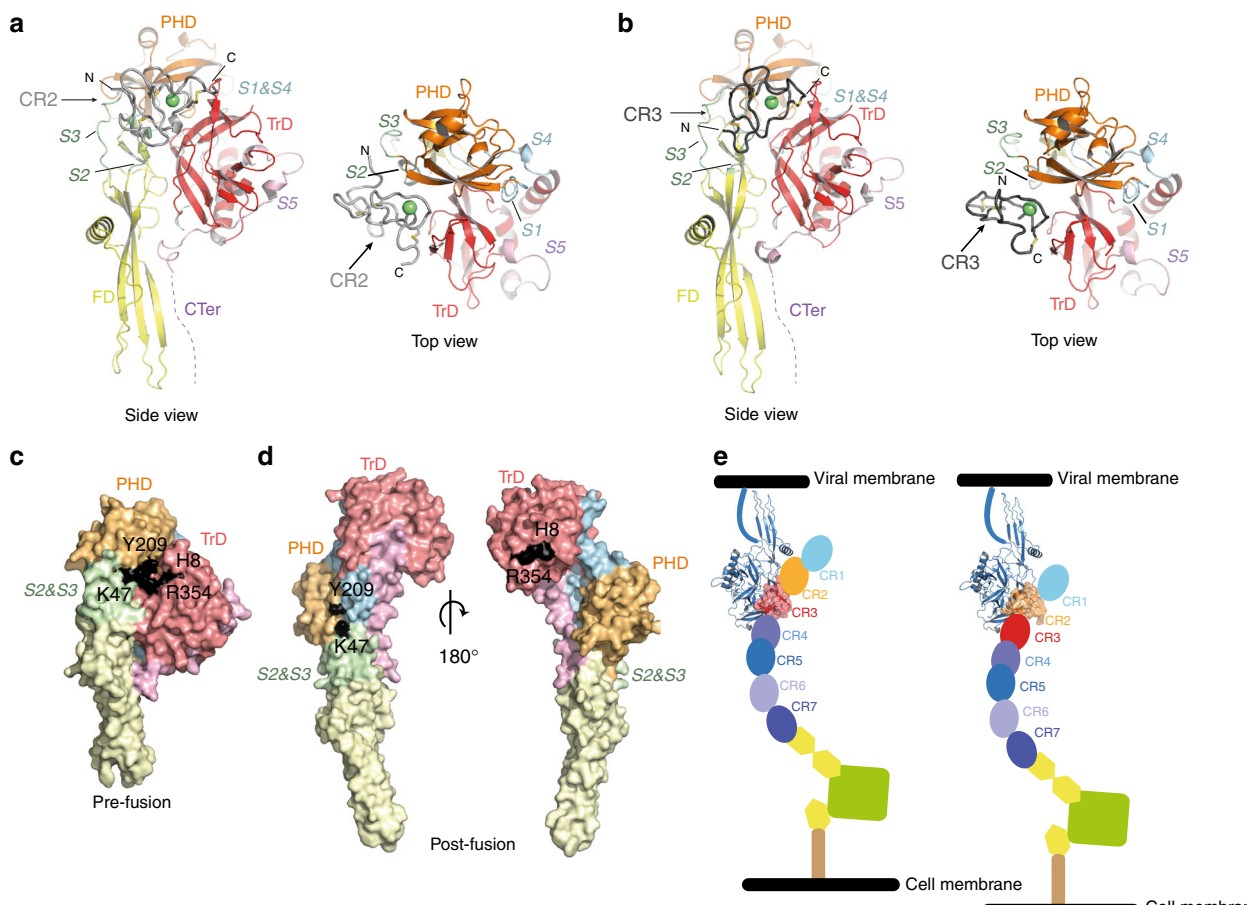

**Fig. 3** X-ray structures of $G_{th}$-CR2 and $G_{th}$-CR3 complexes **a**, **b** Overview of $G_{th}$CR2 (**a**) and $G_{th}$ CR3 (**b**) crystalline structures in ribbon representation. G is depicted by domains and CR domains are in two shades of gray. The conserved disulfides bonds of each CR that maintain their secondary structure are in yellow. In both complexes the CR domain is nested in the same cavity of G. N and C-terminal extremities of each CR are indicated. Color code for $G_{th}$: the trimerization domain (TrD) is in red, the pleckstrin homology domain (PHD) is in orange, the fusion domain (FD) is in yellow. Those domains are connected by segments (S1 to S4) which refold during conformational change: segments S1 and S4 are in cyan, segments S2 and S3 are in green, S5 and the C-terminal segment (CTer) are in purple. The calcium ion of the CR domains is depicted as a green sphere. **c** Footprint of CR2 domain on G pre-fusion conformation. G is in full atoms view and depicted by domains. Residues of G that establish contacts with CR are shown in black on the surface of the protein. **d** Location of residues interacting with CR domains on G post-fusion conformation. Two views at 180° are shown. Note that the interaction patch is scattered when G is in this conformation. **e** Scheme showing the two complexes that can be formed between VSV G and LDL-R. At the cell surface, at neutral pH, the LDL-R adopts an open extended conformation[19] and VSV G can bind either CR2 or CR3. Note that the LDL-R in this extended conformation has the appropriate orientation to interact with G anchored in the viral membrane

composed of a ligand-binding domain, an epidermal growth factor (EGF) precursor homology domain and a C-terminal domain enriched in O-linked oligosaccharides. The ligand binding domain is made of 7 cysteine-rich repeats (CR1 to CR7, Fig. 1a and Supplementary Fig. 1). Each repeat is made of approximately 40 amino acids and contains 6 cysteine residues, engaged in 3 disulfide bridges, and an acidic residues cluster that coordinates a $Ca^{2+}$ ion[17]. The intracellular release of the cargo is driven by a low-pH-induced conformational change of LDL-R from an open to a closed conformation (Supplementary Fig. 1)[17–19].

The LDL-R gene family consists of trans-membrane receptors that reside on the cell-surface, are involved in endocytic uptake of lipoproteins, and require $Ca^{2+}$ for ligand binding. All these receptors have in common several CR repeats (up to several tens), EGF precursor-like repeats, a membrane-spanning region and an intracellular domain containing at least one internalization signal sequence[20]. They are found ubiquitously in all animals including insects[21].

Here we show that VSV G is able to independently bind two distinct CR domains (CR2 and CR3) of LDL-R and we report crystal structures of VSV G in complex with those domains. The structures reveal that the binding sites of CR2 and CR3 on G are identical. We show that HAP-1 cells in which the LDL-R gene has been knocked out are still susceptible to VSV infection confirming that VSV G can use receptors other than LDL-R for entry. However, mutations of basic residues, which are key for interaction with LDL-R CR domains, abolish VSV infectivity in mammalian, as well as insect cells. This indicates that the only receptors of VSV in mammalian and in insect cells are members of the LDL-R family and that VSV G has specifically evolved to interact with their CR domains.

## Results

**LDL-R CR2 and CR3 bind G and neutralize viral infectivity.** We have expressed individually each LDL-R CR domain in fusion with the glutathione S-transferase (GST) in *Escherichia coli*. Proteins were solubilized in the presence of sarkosyl (acting as a

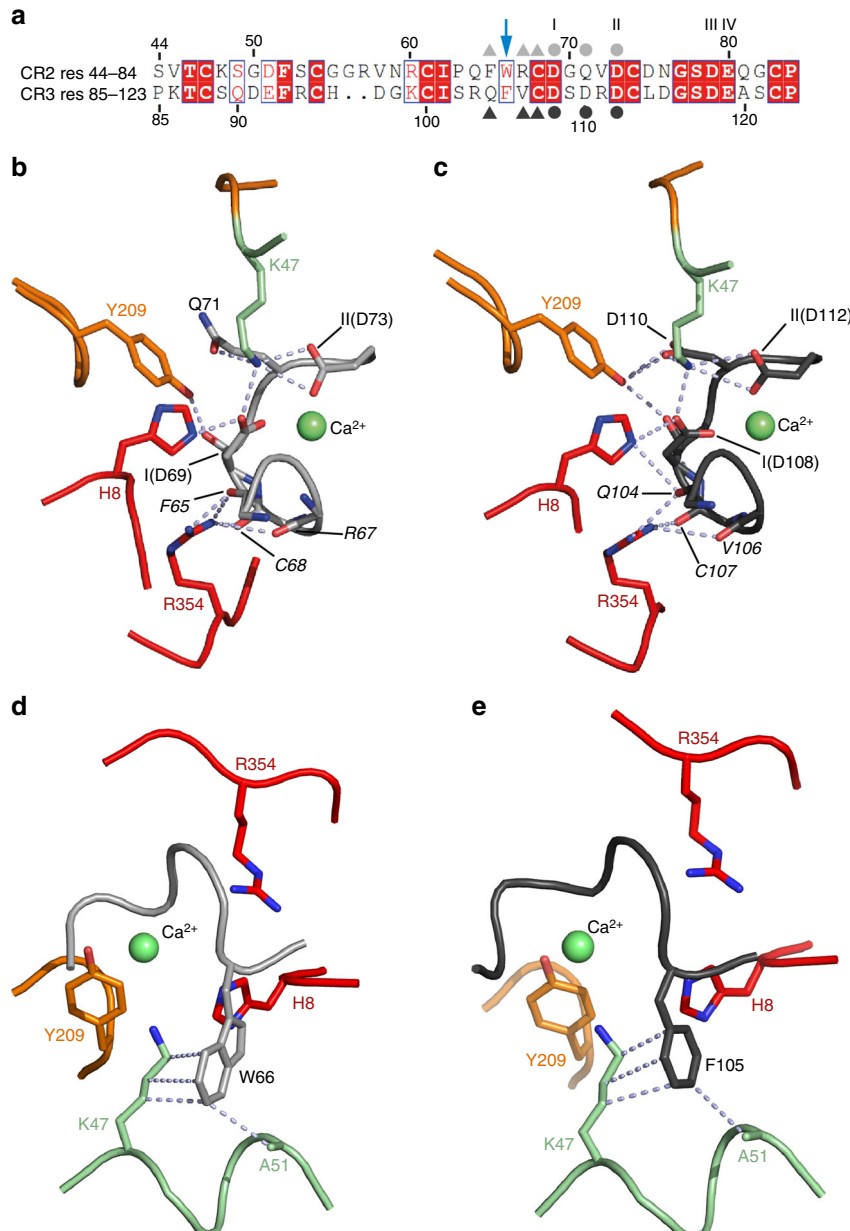

**Fig. 4** Molecular basis of G/CR interaction. **a** Sequence alignment of LDL-R CR2 and CR3. Conserved residues are in a red box and similar residues are shown by red letters boxed in blue. Acidic residues involved in the binding of the $Ca^{2+}$ ion are indicated by I, II, III, and IV. CR residues involved in polar contacts with G are labeled with gray symbols (light gray for CR2 and dark gray for CR3; dots when the contact is established via the lateral chain and triangles when the contact is established via the main chain) on each CR sequence. The aromatic residue which protrudes from the CR modules and establishes hydrophobic interactions with G is indicated by a blue arrow. **b**, **c** Close-up view on the $G_{th}$-CR interface showing the docking of G basic residues on the acidic patch of both CR2 (**b**) and CR3 (**c**). In both cases, the same G residues (H8, K47, Y209, and R354) are involved in the interaction. Residues labels on each CR domain are in italic letters when the contact is established via the main chain; putative bonds are shown as light blue dashed lines. **d**, **e** Close-up view on the $G_{th}$-CR interface showing the hydrophobic interactions between the aromatic residue W66 of CR2 (**d**) and F105 of CR3 (**e**) and residues K47 and A51 of G. The color code is the same as in Fig. 3a and b

mild denaturing agent), DTT and $Ca^{2+}$ and renatured by dilution in a $Ca^{2+}$ containing buffer. The presence of $Ca^{2+}$ was mandatory for correct folding of the proteins. Individual CR domains were then obtained by cleavage of the GST tag by prescission protease. All purified CR domains behave similarly in gel filtration experiments (Supplementary Fig. 2).

Each fusion protein was incubated at pH 8 with magnetic beads coated with glutathione before addition of a soluble form of the ectodomain of G (VSV $G_{th}$, amino acid (AA) residues 1–422, generated by thermolysin limited proteolysis of viral particles[22])

(Fig. 1b). After 20 min of incubation at 4 °C, the beads were washed and the associated proteins were analyzed by SDS/PAGE followed by Coomassie blue staining. This revealed that only CR2 and CR3 domains are able to directly bind VSV G (Fig. 1b) at pH 8. These results are consistent with previous data indicating that a monoclonal antibody (Mab) directed against LDL-R CR3 almost completely inhibited the VSV-triggered cytopathic effect which was not the case with a MAb directed against LDL-R CR6[14]. The binding of $G_{th}$ or VSV to fusion proteins GST-CR2 and GST-CR3 was pH dependent and no interaction was detected at pH 6

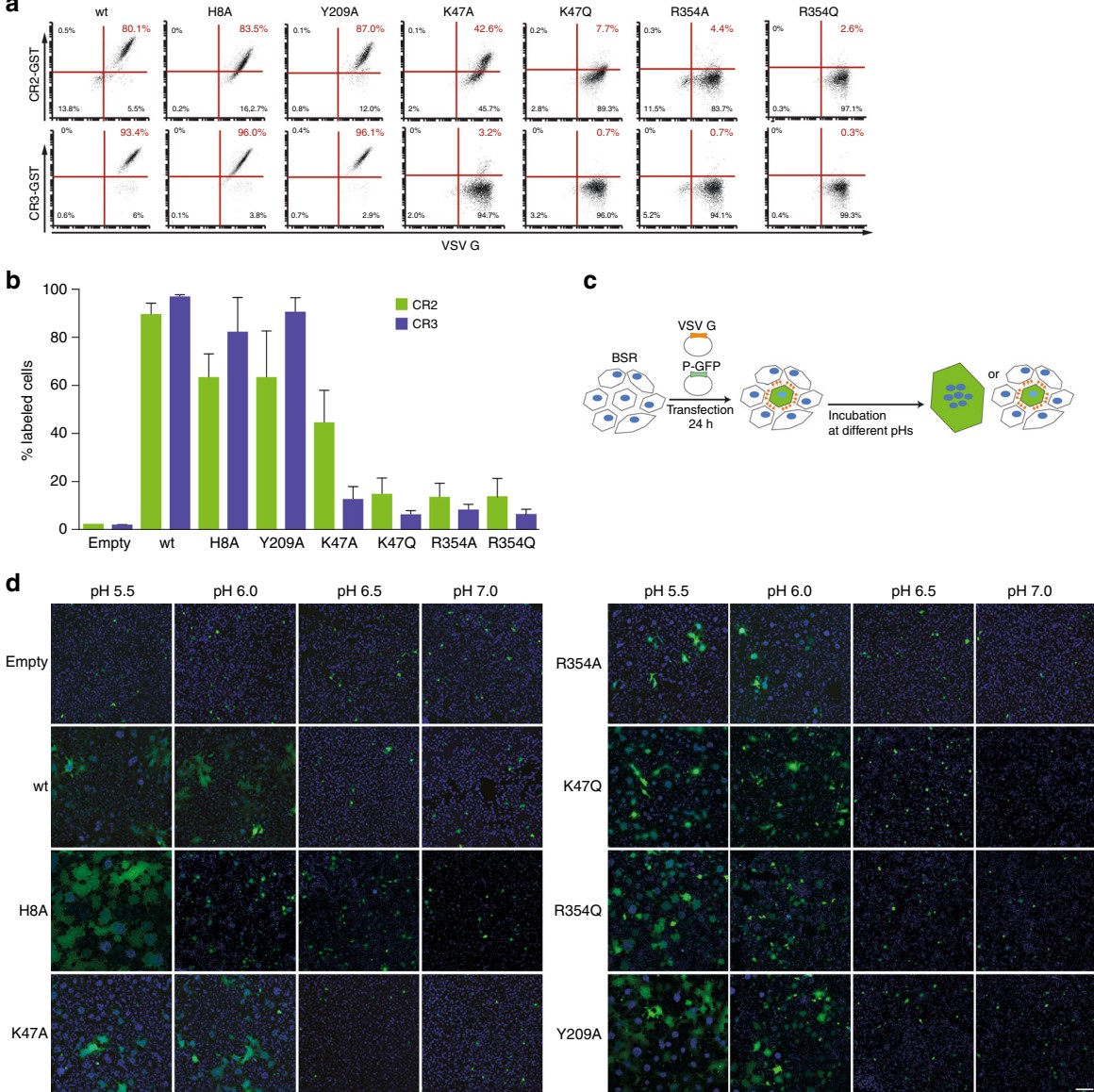

**Fig. 5** K47 and R354 on G are crucial for interaction with CR domains but not required for fusion. **a** Flow cytometry analysis of the expression of WT and mutant glycoproteins at the surface of HEK293T cells and of the binding of fluorescent GST-CR2 (top) and GST-CR3 (bottom). After 24 h of transfection, cell surface expression of WT and mutant G was assessed using monoclonal anti-G antibody 8G5F11 directly on living cells at 4 °C during 1 h. Cells were then incubated simultaneously with anti-mouse Alexa fluor 488 and the indicated GST-CR$^{ATTO550}$. Cells transfected with a G construct that was still able to bind GST-CR proteins exhibited red fluorescence due the ATTO$^{550}$ dye. In each plot, the percentage of ATTO$^{550}$ positive cells is indicated. **b** WT and mutant G ability to bind CR domains. The histogram indicates the mean percentage of ATTO$^{550}$ positive cells for each point mutation on G ($n = 3$). The error bars represent the standard deviation. **c** Cartoon describing the cell–cell fusion assay. BSR cells are co-transfected with plasmids expressing VSV G (either WT or mutant G) and P-GFP. After 24 h of post-transfection cells are exposed for 10 min to media adjusted to the indicated pH which is then replaced by DMEM at pH 7.4. The cells are then kept at 37 °C for 1 h before fixation. Upon fusion, the P-GFP diffuses in the syncytia. **d** Cell–cell fusion assay of WT and mutant glycoproteins. Scale bar=200 μm

(Fig. 1c, d) suggesting that only G pre-fusion conformation is able to bind LDL-R. Finally, GST-CR2 and GST-CR3 (but not GST-CR1) fluorescently labeled with ATTO$^{550}$ (Fig. 1e) specifically recognized VSV G expressed at the surface of infected cells but not the glycoprotein of Chandipura virus (CHAV, another vesiculovirus), which shares 40% AA identity with VSV G (Fig. 1f, g). We also used isothermal titration calorimetry (ITC) to investigate the binding parameters of CR1, CR2, and CR3 to G$_{th}$ in solution (Fig. 2a). Here again, no interaction between G$_{th}$ and CR1 was detected. On the other hand, for both CR2 and CR3, the binding reactions appear to be exothermic, show a

1:1 stoichiometry and exhibit similar $K_d$s in the micromolar range (4.3±1 μM for CR3 and 7.3±1.5 μM for CR2, mean±SEM of three independent experiments).

Recombinant soluble CR2 and CR3 domains, either alone or in fusion with GST, are also able to neutralize viral infectivity when incubated with the viral inoculum prior infection (Fig. 2b). In order to determine the IC$_{50}$ of the different constructions, we incubated $5 \times 10^4$ VSV-eGFP infectious particles with serial dilutions of GST-CR2, GST-CR3, CR2 or CR3. After 15 min, the mixtures were transferred onto $2 \times 10^4$ BSR cells for 30 min of adsorption. After 4 h, the percentage of infected cells (i.e.,

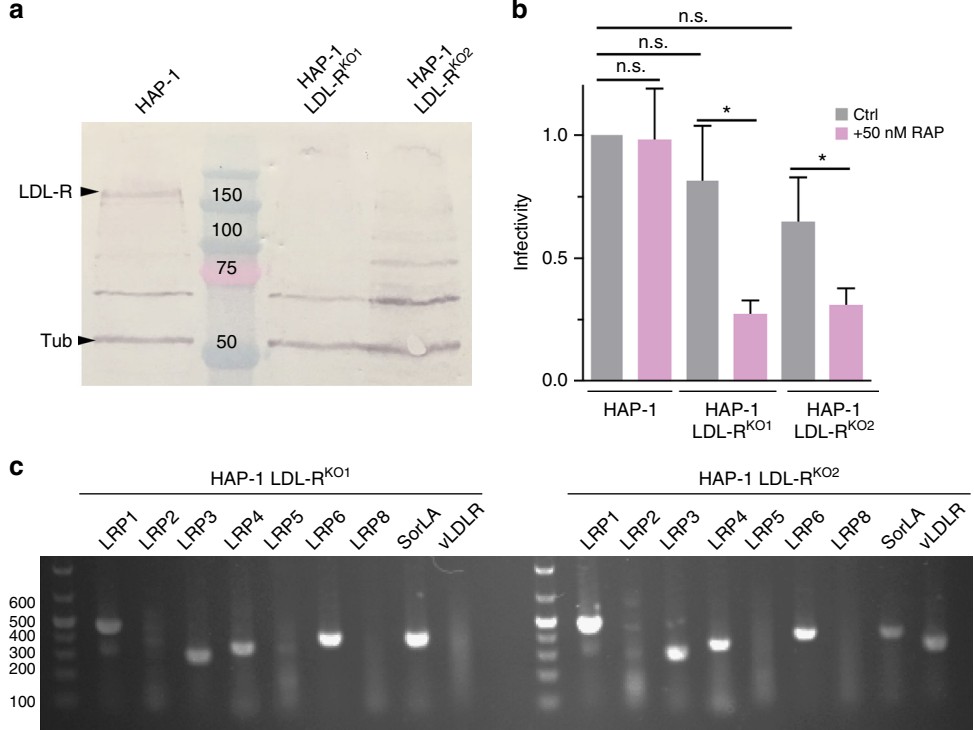

**Fig. 6** LDL-R presence on the cell surface is not mandatory for VSV infection. **a** Analysis of LDL-R expression in wild-type HAP-1, HAP-1 LDL-R[KO1] and HAP-1 LDL-R[KO2] cells. The immunoblot was performed on crude cell extracts and revealed with chicken anti LDL-R (GW22458A from Sigma). α-Tubulin (tub) was also immunoblotted as a loading control. The LDL-R band migrates with an apparent MW of 150 kDa due to the presence of glycosylation sites (predicted MW without oligosaccharides is 95 kDa). **b** Effect of the RAP protein on the susceptibility of LDL-R deficient HAP-1 cells to VSV-eGFP infection. VSV-eGFP was used to infect HAP-1 and HAP-1 LDL-R[KO] cells in the presence or not of RAP (50 nM). Infectivity was determined by counting the number of cells expressing eGFP using a flow cytometer. Data depict the mean with standard deviation for experiments performed in triplicate. $p$ values were determined using an unpaired Student's $t$ test (*$p < 0.05$; n.s. non-significant). **c** Expression of LDL-R family members in HAP-1 LDL-R[KO] cell lines evaluated by RT-PCR

expressing the eGFP) was determined by flow cytometry. For both CR2 and CR3, the $IC_{50}$ is about 15 μM and decreases to about 20 nM when fusion GST-CR constructions are used, thanks to their dimeric nature which induces avidity-enhanced binding to the viral glycoproteins (Fig. 2c).

**Crystal structures of $G_{th}$ in complex with LDL-R CR domains.** We crystallized $G_{th}$ in complex with either CR2 or CR3. The crystal of the complex $G_{th}$-CR3 (diffracting up to 3.6 Å) belongs to space group P622. In this crystal form, the lattice organization of $G_{th}$ molecules is identical to that of the crystal of $G_{th}$ alone in pre-fusion state[8] (Supplementary Fig. 3B). On the other hand, the $G_{th}$-CR2 crystal form (diffracting up to near 2.2 Å) belongs to the H32 space group (Supplementary Fig. 3A).

The binding site of CR domains on G is the same in both crystal forms (Fig. 3a, b). CR2 (resp. CR3) interaction with G buries 1590 Å² (resp. 1450 Å²) of the two molecules' surfaces. It is essentially constituted by segments going from residues 8 to 10 and 350 to 354 in the trimerization domain (TrD), 180 to 184 in the pleckstrin homology domain (PHD) and 47 to 50 in segment S2. Those three segments are rearranged in G post-fusion conformation[7], which explains the inability of CR domains to bind G at low pH (Fig. 3c, d). It is worth noting that the orientation of CR domains in both complexes is optimal for the interaction between G and the open conformation of LDL-R when both proteins are anchored in their respective membranes (Fig. 3e).

Two basic residues of G (H8 from the TrD and K47 from PHD) are pointing toward two acidic residues which belong to

the octahedral calcium cage of the CR domains (D69 and D73 on CR2; D108 and D112 on CR3 labeled I and II—Fig. 4a). The side chain of K47 also establishes an H-bond with the amide group of Q71 (in CR2) and a salt bridge with the acidic group of D110 (in CR3). Two other residues, Y209 (which makes hydrogen bonds with the C=O group of D69 on CR2, and with the C=O group of D108 and the carboxyl group of D110 on CR3) and R354 (which establishes contacts with the main chain of both CR domains), seem also to be key for the interaction (Fig. 4b, c). In both CR domains, the side chain of an aromatic residue (W66 in CR2 and F105 in CR3), which protrudes from the receptor module, also contributes to the stability of the complex by establishing hydrophobic interactions with the aliphatic part of K47 side chain and with the side chain of A51 of G (Fig. 4d, e).

**K47 and R354 of G are crucial for LDL-R CR domains binding.** To investigate their contribution to LDL-R CR domains binding, we mutated residues H8, K47, Y209, and R354 of G into an alanine or a glutamine. HEK cells were transfected with pCAGGS plasmids encoding wild-type or mutant VSV G glycoproteins (WT, H8A, K47A, K47Q, Y209A, R354A, and R354Q). After 24 h of post-transfection, the cells were incubated with MAb 8G5F11 directed against G ectodomain. Then, green fluorescent anti IgG secondary antibodies and GST-CR proteins fluorescently labeled with ATTO[550] were simultaneously added. Immunofluorescence labeling indicated that WT and all G mutants are efficiently transported to the cell surface (Fig. 5a). Mutants H8A and Y209A bind GST-CR proteins as WT G whereas the other mutants are

affected in their binding ability (Fig. 5a, b). Mutants K47Q, R354A and R354Q bind neither GST-CR2 nor GST-CR3. Finally, although no interaction is detected between mutant K47A and CR3, a residual binding activity is observed between this mutant and CR2 (Fig. 5a, b).

We also checked the fusion properties of all the mutants (Fig. 5c). For this, BSR cells were transfected with pCAGGS plasmids encoding wild-type or mutant VSV G glycoproteins. The cells expressing mutant G protein have a fusion phenotype similar to that of WT G (Fig. 5d). This confirms that the mutant glycoproteins are correctly folded and demonstrates

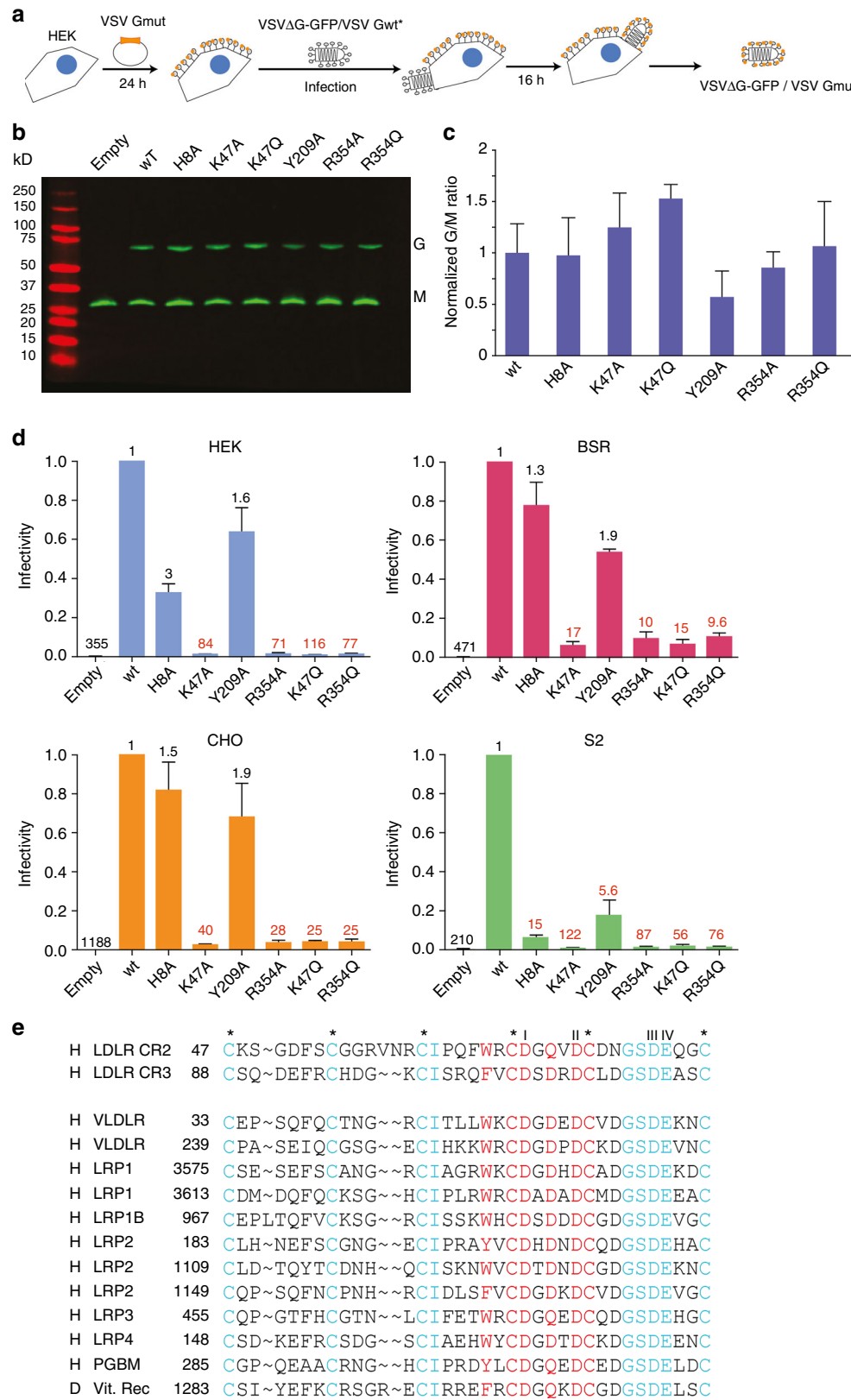

that it is possible to uncouple G fusion activity and receptor recognition.

**Other LDL-R family members are alternative receptors of VSV.** We infected two HAP-1 cell-lines (HAP-1 LDL-R[KO1] and HAP-1 LDL-R[KO2]) in which the LDL-R gene has been knocked out (Fig. 6a and Supplementary Fig. 4). Both were as susceptible to VSV infection as WT HAP-1 cells (Fig. 6b). This demonstrates that VSV receptors other than the LDL-R are present at the surface of HAP-1 cells.

To evaluate the role of other LDL-R family members[23] as VSV receptors, we took advantage of the properties of the receptor-associated protein (RAP), a common ligand of all LDL-R family members[24–26] which blocks ligand binding to all LDL-R family members with the exception of LDL-R itself[14,26]. RAP significantly inhibits VSV infection in HAP-1 LDL-R[KO] cell lines but not in WT HAP-1 cells (Fig. 6b). Those results are consistent with previous data suggesting that VSV can use other LDL-R family members as alternative receptors[14]. Indeed, RT-PCR on purified RNA revealed the expression of several LDL-R family members including LRP1, LRP3, LRP4, LRP6, and sortilin (SorLA) in both HAP-1 LDL-R[KO] cell lines. Messengers corresponding to the vLDL-R were only detected in HAP-1 LDL-R[KO2] (Fig. 6c).

**G mutants affected in CR binding cannot rescue VSVΔG-GFP.** We then examined whether the mutant glycoproteins described above are able to sustain viral infection. We used a recombinant VSV (VSVΔG-GFP) in which the G envelope gene was replaced by the green fluorescent protein (GFP) gene and which was pseudotyped with the VSV G glycoprotein[27,28]. This pseudotyped recombinant was used to infect HEK cells either transfected or not transfected by a plasmid encoding WT or mutant glycoproteins[29]. After 16 h, the infected cells supernatant was collected (Fig. 7a). Mutant glycoproteins incorporation into the envelope of the particles present in the supernatant was verified by western blot (Fig. 7b, c) and the infectivity of the pseudotyped particles was analyzed in different cell lines (mammalian HEK, BSR, CHO and Drosophila S2 cells) by counting the cells expressing GFP by flow cytometry 4 h p.i. (Fig. 7d). Mutants K47A, K47Q, R354A, and R354Q did not rescue the infectivity of VSVΔG-GFP. Compared to WT G, the infectivity decreased by a factor of 10 up to 120 (Fig. 7d). The decrease was more important in HEK and S2 cell lines than in the two hamster cell lines. In mammalian cell lines, mutants H8A and Y209A can rescue the infectivity of VSVΔG-GFP at a level similar to that of WT. This is not the case in S2 cell line, where their infectivity significantly decreased (by a factor of 15 for mutant H8A and ~6 for Y209A) (Fig. 7d).

As the fusion activity of the mutants is unaffected (Fig. 5d), the loss of infectivity of pseudotypes bearing a mutant glycoprotein can be safely attributed to their disability to recognize a cellular receptor. These results indicate that mutants K47A, K47Q, R354A, and R354Q which are unable to bind LDL-R CR domains are also severely impaired in their ability to bind other VSV receptors.

**Discussion**

LDL-R has been demonstrated to be the major entry port of VSV and lentivirus pseudotyped by VSV G[14]. Here, we demonstrate that VSV G is able to bind two CR domains of the LDL-R with similar affinities. The biological relevance of this interaction was demonstrated by the ability of both CR2 and CR3 to inhibit VSV infection. The crystal structures of VSV G in complex with CR2 and CR3 reveal that they both occupy the same site at the surface of the glycoprotein in its pre-fusion conformation and that the same G residues ensure the correct anchoring of the CR domains. This binding site is split apart when G is in its post-fusion conformation, which explains why G is unable to bind CR domains at low pH. This may disrupt the interaction between G and LDL-R in the acidic endosomal lumen and favor the transport of the virion to an appropriate fusion site.

CR domain recognition by VSV G involves basic residues H8 and K47, pointing toward the calcium-coordinating acidic residues, and R354 which interacts with C=O groups of the main chain. This mode of binding is very similar to what is observed for endogenous ligand recognition by CR domains of the LDL-R family members[17,30,31] and, indeed, mutant glycoproteins in which either K47 or R354 is replaced by an alanine or a glutamine, are unable to bind CR domains. It is worth noting that those key residues are not conserved among vesiculoviruses. Therefore, the use of LDL-R as a viral receptor cannot be generalized to the other members of the genus. Indeed, we have shown that CHAV G, which does not possess basic residues in positions corresponding to VSV residues 47 and 354, does not bind CR domains.

VSV G binds only CR2 and CR3. All CR domains have the same fold and all form a calcium cage involving conserved acidic residues. The acidic residues in position I and II, which play a key role in the interaction with G, are conserved except in CR7 which has an asparagine in position I (Supplementary Fig. 1). Therefore, they are poorly discriminant for the interaction between a given CR domain and G. Similarly, residue R354 of G interacts with the main chain of both CR2 and CR3 and the influence of CR domains amino acid sequence on this interaction is difficult to predict. However, the aromatic residues (W66 in CR2 and F105 in CR3) establishing hydrophobic interactions with the aliphatic

**Fig. 7** Infectivity of VSVΔG-GFP virus pseudotyped with mutated G in various cell types. **a** Generation of VSVΔG-GFP virus pseudotyped with VSV G mutants. Transfected HEK-293T expressing mutant G were infected with VSVΔG-GFP pseudotyped with VSV G WT. After 16 h, VSVΔG-GFP virions pseudotyped with mutant VSV G were harvested. **b, c** Incorporation of WT and mutant G in VSVΔG-GFP viral particles. VSVΔG-GFP pseudotyped with the WT VSV G was used to infect HEK-293T (MOI=1) transfected with indicated mutants. After 16 h, viral supernatants were analyzed by Western blot (using an anti-VSV G and an anti-VSV M antibody) (**b**). For each mutant, the G/M ratio was compared to that of WT normalized to 1. The mean of 3 independent experiments is shown with error bars representing the SD (**c**). **d** Infectivity of VSVΔG-GFP pseudotyped with WT and mutant glycoproteins. VSVΔG-GFP pseudotyped with WT VSV G was used to infect HEK-293T previously transfected with the indicated mutated glycoprotein (MOI 1). VSVΔG-GFP viruses pseudotyped by WT or mutant glycoproteins were used to infect indicated cell lines during 4 h; the percentage of infected cells was determined by counting GFP expressing cells by flow cytometry. Data depict the mean with SD for three independent experiments. Above each bar, the reduction factor of the titer (compared to VSVΔG-GFP pseudotyped by WT G which was normalized to 1) is indicated. **e** Sequence alignment of human (h) and drosophila (d) CR domains having characteristics similar to human LDL-R CR2 and CR3. Besides scaffolding residues of the CR fold (i.e., the 6 cysteines, labeled by stars, and the four acidic residues—I, II, III, IV—forming the calcium octahedral cage), we imposed (i) the absence of insertion or deletion in the contact region, (ii) an aromatic residue corresponding to W66 of CR2, and (iii) an amidic or an acidic amino acid in place corresponding to residue Q71 of CR2. PGBM: Basement membrane-specific heparan sulfate proteoglycan core protein. Vit.Rec: Putative vitellogenin receptor. Numbering of residues corresponds to that of the mature protein for CR2 and CR3 and to the precursor in all other instances (i.e., including the signal peptide)

part of K47 side chain and with the side chain of A51 of G are not conserved and are replaced by an arginine in CR6 and a lysine in CR7 (Supplementary Fig. 1). Also, instead of an amide or an acidic amino acid in place corresponding to residue Q71 in CR2 and D110 in CR3 (which interacts with the side chain of K47), a serine is found in CR1, a glycine in CR5 and an alanine in CR7 (Supplementary Fig. 1). All those considerations may explain why we do not observe G binding to CR1, CR5, CR6 or CR7. The question remains opened for CR4.

Our functional analysis confirms that LDL-R is not the only receptor of VSV[14] as HAP-1 LDL-R$^{KO}$ can be infected as efficiently as HAP-1 cells. However, the mutant glycoproteins which are unable to bind CR domains cannot restore VSVΔG-GFP infectivity neither in mammalian nor in insect cells. The most parsimonious interpretation of this result is that the only receptors of VSV are members of the LDL-R family or other transmembrane proteins containing CR domains. The molecular basis of the interaction is the same for all those receptors and involves G ability to bind their CR domains. This is in agreement with the decrease of infectivity observed in presence of RAP protein which is an antagonist of other members of the LDL-R family[26].

Indeed, a systematic search of CR domains having similar characteristics to CR2 and CR3 (i.e. besides the common characteristics of the CR domains, they must have no insertion or deletion in the contact region, an aromatic residue corresponding to W66 of CR2 to interact with the aliphatic part of K47, and an amide or an acidic amino acid in place corresponding to residue Q71 of CR2) reveals two such CR domains in VLDL-R, two in LRP1, one in LRP1B, three in LRP2, one in LRP3, one in LRP4, one in PGBM (Basement membrane-specific heparan sulfate proteoglycan core protein) in humans. In Drosophila, we found a single CR domain with those characteristics in the putative vitellogenin receptor (Fig. 7e).

Overall this study demonstrates that VSV G has specifically evolved to interact with CR domains of the members of the LDL-R family. The ubiquitous nature of this receptor family (which is also widespread among invertebrates) explains the pantropism of VSV. Other proteins from other families (such as the corin, which is a transmembrane serine protease[32]) also contain CR domains. However, their CR domains do not seem to have the correct characteristics to interact with G (and are not retrieved in our systematic search). Furthermore, corin is a type II membrane protein imposing an opposite orientation of CR domains, a feature which may also impede G binding.

The LDL-R family members are involved in entry of several viruses belonging to different families. Among them are some human rhinoviruses (HRV)[33] and hepatitis C virus[34,35]. The crystal structure of a complex between the third CR module of the very low-density lipoprotein receptor and HRV2 reveals that the attachment site on the virus involves a conserved lysine residue pointing toward the calcium cage[36]. Similarly to VSV G, the side chain of this lysine establishes hydrophobic interactions with a tryptophan residue of the CR module. This tryptophan corresponds to W66 on CR2 and F105 on CR3 of the LDL-R receptor (Fig. 4d, e). Therefore, the molecular basis of CR module recognition by HRV2 VP1 protein and VSV G are very similar. Furthermore, in the case of HRV2, the crystal structure and experimental data also indicated that different CR domains and different members of the family redundantly play the role of HRV2 receptors[33,36].

The demonstration that the receptors of VSV are all members of the LDL-R family together with the characterization of the molecular basis of CR domains recognition by G paves the way to develop recombinant VSVs with modified tropism. Indeed, a glycoprotein having (i) a point mutation which ablates the natural receptor tropism and (ii) an insertion of a protein domain or a peptide targeting specifically a tumor cell[37] should allow the design of fully retargeted oncolytic VSVs. Such viruses should be able to eliminate cancerous cells while sparing normal ones.

## Methods

**Cells and viruses**. BSR, a clone of BHK-21 (Baby Hamster Kidney cells; ATCC CCL-10) and HEK-293T (human embryonic kidney cells expressing simian virus 40T antigen; ATCC CRL-3216) cells were grown in Dulbecco's modified Eagle's medium (DMEM) supplemented with 10% fetal calf serum (FCS). HAP-1 wt and HAP-1 LDL-R deficient cells (HAP-1 LDL-R$^{KO1}$—catalog number: HZGHC003978c007—and HAP-1 LDL-R$^{KO2}$—catalog number: HZGHC003978c008—purchased from Horizon Discovery) were grown in Iscove's Modified Dulbecco's Medium (IMDM) supplemented with 10% FCS. CHO (cell line derived from Chinese hamster ovaries) cells were grown in Ham's F12 medium supplemented with 2 mM glutamine and 10% FCS. All mammalian cell lines were maintained at 37 °C in a humidified incubator with 5% CO$_2$. Drosophila S2 cells were grown in Schneider's medium supplemented with 10% FCS.

Wild-type VSV (Mudd-Summer strain, Indiana serotype), VSVΔG-GCHAV[38] and VSV-eGFP (a gift from Denis Gerlier) were propagated in BSR cells.

VSVΔG-GFP is a recombinant VSV which was derived from a full-length cDNA clone of the VSV genome (Indiana serotype) in which the coding region of the G protein was replaced by a modified version of the GFP gene and pseudotyped with the VSV G protein[29]. VSVΔG-GFP was propagated on HEK-293T cells that had been previously transfected with pCAGGS-VSVG.

**Plasmids and cloning**. Point mutations were created starting from the cloned VSV G gene (Indiana Mudd-Summer strain) in the pCAGGS plasmid. Briefly, forward and reverse primers containing the desired mutation were combined separately with one of the primers flanking the G gene to generate two PCR products. These two G gene fragments overlap in the region containing the mutation and were assembled into the pCAGGS linearized vector using Gibson assembly reaction kit (New England Biolabs).

**Antibodies**. Chicken anti LDL-R (GW22458A, dilution: 1/500) and α-Tubulin (T6199, dilution 1/1000) were purchased from Sigma. Anti G (8G5F11, dilution 1/1000) was purchased from Kerafast. Goat anti-mouse Alexa fluor 488, purchased from ThermoFisher (A-11029, dilution 1/1000), were used as secondary antibodies. Goat anti-rabbit DyLight 800 conjugate (from Cell Signaling Technology, ref 5251, dilution 1/1000) was also used as a secondary antibody in western blot.

**Protein expression, purification, and labeling**. VSV G$_{th}$ was obtained by limited proteolysis of viral particles and purified, as previously described[22]. Briefly, 4 volumes of concentrated virus (around 10 mg/ml) was thawed and incubated for 30 min at 37 °C with one volume of phosphate-citrate buffer at pH 6.3. A 1 mg/ml solution of thermolysin was applied directly on viral particles (with a ratio viral solution/thermolysin solution of 10/1 v/v) during 1 h at 37 °C for limited proteolysis. The reaction was stopped by addition of blocking buffer (900 mM Tris-HCl pH 8.8, 50 mM EDTA, supplemented with a cocktail of protease inhibitors from Roche). Then, the soluble G ectodomain (G$_{th}$) was separated from the "shaved" viral particle by centrifugation at 13,000 g during 45 min. The supernatant was loaded on the top of a 20% sucrose, 20 mM Tris-HCl pH 8.8 cushion and spun for 50 min at 40,000 rpm in a SW55 rotor. The supernatant was diluted 10 times in Tris–HCl 10 mM pH 8.8 and loaded on a DEAE trisacryl column. G$_{th}$ was eluted with 200 mM NaCl, 20 mM Tris-HCl pH 8.8 buffer. The eluted G$_{th}$ was injected on a Superdex 200 HR10/30 column and eluted in 20 mM Tris-HCl pH 8.8 buffer. Fractions containing the G$_{th}$ were pooled and concentrated with an amicon C/O 30 kDa ultrafiltration unit (Millipore).

As for the LDL-R, DNA sequences encoding the 7 CR domains of the human LDL-R (NM_000527, GenBank) were synthetized (MWG biotech) and subcloned in the pGEX-6P1 bacterial expression vector (Invitrogen). Each protein construct contains at its N-terminus a GST tag and a preScission protease cleavage site. Each CR domain was purified using the following protocol derived from ref. [39]. C41 bacteria transformed with the CR construct were cultured at 37 °C in LB-ampicillin medium until OD reached 0.6 AU. Protein expression was then induced with 1 mM IPTG during 5 h at 37 °C. Cells were sonicated in lysis buffer (500 mM NaCl, 20 mM Tris–HCl pH 8, 2 mM CaCl$_2$, 2% w/v sarkosyl and 1 mM DTT). The clarified supernatant was incubated with glutathione agarose beads (Thermo Fisher Scientific) in presence of 0.2% Triton X100 during 2 h. After incubation, beads were then extensively washed with equilibration buffer (200 mM NaCl, 20 mM Tris HCl pH 8, 2 mM CaCl$_2$, 1 mM PMSF). The GST-CR construct was then eluted with the same buffer supplemented with 20 mM GSH. Purification of each GST-CR was achieved with a gel filtration step using a Superdex 200 column (Ge Healthcare). To isolate CR domains, purified GST-CR was incubated with preScission protease and injected on a gel filtration column Superdex 75 (Ge Healthcare). Fractions containing pure CR domains were then pooled, concentrated at 1 mM and stored at −80 °C until use.

One milligram of purified GST-CR2 (or GST-CR3) was labeled with the fluorescent dye ATTO$^{550}$ NHS ester (Sigma Aldrich) using the instruction of the

manufacturer. The labeled proteins were then diluted at a concentration of 50 μM and stored at −80 °C until use. The labeling ratio was estimated to be around 2 dyes per molecule.

**Crystallization and structure determination**. Initial crystallization screen was performed at 20 °C by sitting drop vapor diffusion (100 nl protein and 100 nl mother liquor) in 96-well TTP plates (Corning) using a Mosquito robot.

Crystals of G$_{th}$ in complex with CR2 domain were grown in hanging drops containing equal volumes (1 μL) of purified proteins and reservoir solution (29% PEG 3000, 100 mM Tris–HCl pH 8.5, 200 mM LiSO$_4$). Crystals were observed after 2–3 days and matured to full size within a week. Crystals were soaked into mother liquor supplemented with 35% (v/v) glycerol for cryo-protection. Crystals were plunged into liquid nitrogen and measured at the SOLEIL synchrotron beam line PROXIMA-1. A data set was collected at 100 K on a single G$_{th}$-CR2 crystal to near 2.3 Å resolution.

Crystals of G$_{th}$ in complex with CR3 domain were grown in hanging drops containing equal volumes (1 μL) of purified proteins and reservoir solution (30% PEG 400, 50 mM Tris–HCl pH 8.5, 200 mM CaCl$_2$, 0.2% dodecylmaltoside). Crystals were observed after 1–2 days and cryo-protected directly on the cryo stream at the SOLEIL synchrotron microfocus beam line PROXIMA-2. A full data set was collected at 100 K on a single G$_{th}$-CR3 crystal to 3.6 Å resolution.

**Structure determination and refinement**. All diffraction data were integrated and reduced using XDS program package[40]. The crystals of the G$_{th}$-CR2 complex belonged to H32 space group and those of the G$_{th}$-CR3 complex to the P622 space group. In both cases, the structures were determined by molecular replacement with MOLREP[41] using G$_{th}$ in its pre-fusion conformation as search model (PDB 5I2S). Initial maps were clear enough to assess the presence of the CR domain in the crystals. As crystals of G$_{th}$-CR2 were anisotropic, data were submitted to the STARANISO server (http://staraniso.globalphasing.org/cgi-bin/staraniso.cgi) to apply an anisotropy correction of the data. These corrected data were then used for the refinement of the G$_{th}$-CR2 structure. For both structures, the model was iteratively built using COOT[42] and refined with AutoBUSTER[43] and Phenix[44]. The details of the crystallographic analysis are presented in Supplementary Table 1. Structure representations were made using PyMol (PyMOL Molecular Graphics System. DeLano Scientific LLC, San Carlos, CA, USA. http://www.pymol.org). Surfaces interactions were computed with the PISA server (http://www.ebi.ac.uk/msd-srv/prot_int/cgi-bin/piserver).

**Characterization of the binding between G and CR domains**. Purified GST-CR proteins were incubated with magnetic beads coated with GSH (Eurogentec) under agitation during 20 min at 4 °C. Then, the slurry was washed with the equilibration buffer at the appropriate pH (200 mM NaCl, 2 mM CaCl$_2$, 50 mM Tris-HCl pH 8 or 50 mM MES-NaOH pH 6). Purified G$_{th}$ or viral particles were preincubated in this same buffer for 20 min and added to the magnetic beads bound to GST-CR construction or GST alone. After 20 min of incubation under soft agitation, the slurry was washed two times with the equilibration buffer at the appropriate pH (either 8 or 6). Beads were re-suspended in the gel loading buffer and directly analyzed on a SDS PAGE.

**ITC**. ITC experiments were performed at 293 K using a MicroCal iTC200 apparatus (GE Healthcare) in a buffer composed of 200 mM NaCl, 20 mM Tris–HCl pH 8.0 and 2 mM CaCl$_2$. G$_{th}$, at a concentration of 50 μM, was titrated by successive injections of CR domains at a concentration of 600 μM. The titration sequence included a first 1 μL injection followed by 19 injections of 2 μL each with a spacing of 180 or 240 s between injections. OriginLab software (GE Healthcare) was used to analyze the raw data. Binding parameters were extracted from curve fitting analysis with a single-site binding model.

**Binding of CR domains to cells expressing WT or mutant G**. For microscopy, BSR cells were infected for 4 h and were then incubated with GST-CR2$^{ATTO550}$ or GST-CR3$^{ATTO550}$ at 4 °C for 30 min. Cells were fixed with 4% paraformaldehyde and then permeabilized with 0.5% Triton X-100. Nucleoprotein was detected by using a mouse monoclonal anti-VSV N antibody (homemade) at a dilution of 1/1000. Goat anti-mouse Alexa fluor 488 was used as secondary antibody. Images were captured using a Leica SP8 confocal microscope (×63 oil-immersion objective).

For flow cytometry experiments, HEK-293T cells were transfected with pCAGGS plasmids encoding WT or mutant G using polyethylenimine (PEI, Sigma-Aldrich). 24 h after transfection, cells were collected and incubated with a mouse-monoclonal anti-G antibody that recognizes G ectodomain (8G5F11, KeraFast) at a dilution of 1/1000. Goat anti-mouse Alexa fluor 488 (at a dilution of 1/1000) and GST-CR2$^{ATTO550}$ (or GST-CR3$^{ATTO550}$) at a concentration of 0.5 μM were then simultaneously added to the cells. The fluorescence of cells was determined using a BD Accuri C6 flow cytometer.

**Neutralization assay**. VSV-eGFP was incubated with increasing concentrations of CR domains (either alone or in fusion with GST) for 30 min. The virus-CR

domains mixtures were then used to infect BSR cells, plated on 96-well plates at 70 % confluence (in the absence of CR domains the MOI of the inoculum was 1). The percentage of infected cells (eGFP-positive) was determined 4 h p.i. using a BD Accuri C6 flow cytometer.

**HAP-1 cells infection**. HAP-1 cells were plated at 70% confluence and incubated, or not, with 50 nM of RAP (Enzo life science) during 15 min. Cells were then infected with VSV-eGFP at an MOI of ~1. RAP was maintained during all the infection time. The percentage of infected cells (GFP-positive) was determined 4 h p.i. using a BD Accuri C6 flow cytometer. Experimental MOIs were precisely determined using the equation $MOI = -\ln[p(0)]$ where $p(0)$ is the proportion of non-infected cells and compared to that of VSV-eGFP in untreated HAP-1 cells which was normalized to 1.

**Pseudotypes**. HEK-293T cells at 80% confluence were transfected by pCAGGS encoding WT or mutant VSV G using PEI. At 24 h after transfection, cells were infected with VSVΔG-GFP at an MOI of 1. Two hours p.i., cells were washed to remove residual viruses from the inoculum. Cell supernatants containing the pseudotyped viral particles were collected at 16 h p.i. The infectious titers of the pseudotyped viruses were determined on non-transfected cells by counting cells expressing the GFP using a BD Accuri C6 flow cytometer at 4 h p.i. WT and mutant G incorporation in the pseudotyped particles was assessed after supernatant concentration by SDS PAGE and western blot analysis using a rabbit-polyclonal anti-VSV G (1/5000) and a rabbit-polyclonal anti-VSV M (both homemade) (1/5000) and a goat anti-rabbit DyLight 800 conjugate was used as secondary antibody. The blot image was obtained using a LI-COR Biosciences Odyssey®. Experimental MOIs were precisely determined using the equation $MOI = -\ln[p(0)]$ where $p(0)$ is the proportion of non-infected cells and compared to that of VSVΔG-GFP pseudotyped with WT G which was normalized to 1.

**Cell–cell fusion assay**. Cell–cell fusion assay was performed, as previously described[10,29] (Fig. 5c). Briefly, BSR cells plated on glass coverslips at 70% confluence were co-transfected with pCAGGS plasmids encoding WT or mutant G, and P-GFP plasmid encoding the phosphoprotein of Rabies virus fused to GFP. After 24 h of transfection, cells were incubated with fusion buffer (DMEM-10mM MES) at various pHs (from 5.0 to 7.5) for 10 min at 37°. Cells were then washed once and incubated with DMEM-10mM HEPES-NaOH buffered at pH 7.4, 1% BSA at 37 °C for 1 h. Cells were fixed with 4% paraformaldehyde in 1× PBS for 15 min. Cells nuclei were stained with DAPI and syncytia formation was analyzed with Zeiss Axiovert 200 fluorescence microscope with a ×10 lens.

**RT-PCR**. Total RNA from HAP-1 cells was extracted using TRIzol (Invitrogen) and Direct-zol RNA Kit (Zymo Research) according to the manufacturer's protocol. Primer sequences were chosen spanning an exon-exon junction and are described in Supplementary Table 2. cDNA synthesis and PCR amplification steps were performed in a single reaction using the OneTaq One-Step RT-PCR Kit (Biolabs).

**Data availability**. The atomic coordinates and structure factors have been deposited in the Protein Data Bank (accession code 5OYL and 5OY9). All other relevant data are available from the authors.

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

## Acknowledgements

This work was supported by grants from the Fondation pour la Recherche Médicale (FRM DEQ20120323711) and by grants from ANR (ANR 11 BSV8 002 01 and ANR CE11 MOBARHE) to Y.G. We acknowledge the European Synchrotron Radiation Facility (Grenoble, France) and synchrotron SOLEIL (Saint-Aubin, France) for provision of radiation facilities. Access to crystallization platform was supported by the French Infrastructure for Integrated Structural Biology (FRISBI) ANR-10-INSB-05. This work has benefited from the expertize of the Macromolecular Interactions Measurements Platform of I2BC. We particularly thank Magali Aumont for the technical assistance and Michel Desmadril for his expertize in ITC. We also thank Danielle Blondel, Abbas Abou-Hamdan and Cécile Lagaudrière-Gesbert for careful reading of the manuscript.

## Author contributions

J.N. designed, performed and analyzed cellular virology experiments; L.B. purified and characterized the proteins, grew the crystals, participated in diffraction data collection, in data processing and in the structure determination, and performed and analyzed cellular virology experiments; P.L. participated in diffraction data collection, in data processing and in the structure determination and refined the structure; H.R. performed cell–cell fusion assays. Y.G. supervised the work, designed the experiments, analyzed all the data and wrote the manuscript; A.A. designed and performed the experiments, analyzed all the data and wrote the manuscript.

## Additional information

**Competing interests:** The mutants described in this work are the subject of a patent application by CNRS on which J.N., L.B., H.R., A.A. and Y.G. are named as inventors. P.L. declares no competing interests.

