## [Peer Review File · Nature Communications]

Reviewers' comments:

Reviewer #1 (Remarks to the Author):

The current study investigates the structural basis for recognition of VSV glycoprotein by LDL receptor family members. This is an important and significant study for two reasons: First, the study adds to our knowledge of ligand recognition by this family of receptors, and second, provides structural insight into how vesicular stomatitis virus interacts with host cell receptors. Overall, the study appears to be carefully done. However, attention to some minor details would improve the study significantly.

1. The CR repeats from the LDL receptor were prepared in bacteria as a fusion protein with GST. Were these fragments subjected to a refolding process, which would be necessary to obtain correctly folded domains? This is particularly important for the data generated from Figure 2.
2. The binding of VSV glycoprotein G to CR2 and CR3 is understandably weak (K_d values in the μM range). However, the binding of VSV to cellular receptors presumably occurs with much higher affinity. How does this occur? Is it a result of multivalent interactions? The authors should discuss this and perhaps revise the model depicted in Fig 3e to describe this more precisely.
3. The authors conclude from the experiments depicted in Figure 6 that other LDL receptor family members participate as receptors for VSV. It would be of interest to determine which LDL receptor family members are expressed in these HAP-1 cells. This could be done by PCR for example.
4. In the experiments shown in Figure 6B, the RAP concentration should be given.

Reviewer #2 (Remarks to the Author):

Albertini and colleagues report the crystal structures of VSV G in complex with two cysteine-rich domains (CR2 and CR3) of the LDL receptor. Both CR2 and CR3 bind to the same site on VSV G. Mutational analyses of the G binding site shows that VSV loses its infectivity, thus indicating that LDL-R constitutes an essential receptor for VSV entry.

The main results are:

- CR2 and CR3 binding to G is pH dependent
- CR2 and CR3 bind G in the low micromolar range
- CR2 and CR3 domains block VSV G infection in a cellular model
- The co-crystal structures define the interaction interface with contributions from TRD, PHD and S2. These regions do not form a continuous surface in low pH G
- Two basic residues of G have been identified as key residues for interaction
- Knock-down of LDL-R indicates that HAP-1 cells can be still infected with VSV. The authors conclude that alternative receptors may allow infection. However, the western blot of fig. 6 shows a faint band in the knock-down cells and this residual LDL-R expression may be sufficient for infection. In addition, RAP only blocked VSV infection to 50% in the knockdown cells. This should be considered in the discussion of alternative receptors.

G proteins carrying mutations of the basic residues K47 and R354 do not rescue VSV infection, corroborating the importance of an intact receptor interaction surface.

In summary, the manuscript is well written (some editing required) and of high technical quality.

The results are presented and discussed in a clear way and they support the major conclusions.

The work thus represents an important step forward in understanding the molecular details of VSV

G interaction with its cellular receptors.

Minor comments:

The R.m.s. deviation of the bond angles (1.8°) is rather high in the 2.25 Å structure!

The line indicating the resolution range of the processed data needs to be shifted in the table

Reviewer #3 (Remarks to the Author):

This manuscript reports structural and functional studies of the VSV G protein with the LDL receptor. The LDL receptor is a receptor for VSV entry and interacts directly with the VSV G protein. Here the authors dissect the interactions of the LDL receptor with VSV G, by expressing individual CR domains. Only CR2 and CR3 are shown to interact with prefusion G and crystal structures of both complexes are obtained. Based on the crystal structure analysis, residues of G which interact with the calcium binding site on the CR domains are implicated in receptor binding and entry. Mutagenesis of G demonstrates that these interactions are critical for LDL-R binding as well as more generally for proper G function in entry. These data indicate that other receptors that mediate VSV entry form similar interactions and implicate other LDL-R family members as potential receptors. Although overall the manuscript is well written and the data is convincing, there are a number of issues with the current version that need to be addressed.

- Individual CR domains from the LDL-R are expressed for analysis and the data indicate that only CR2 and CR3 interact with G. However, there is no independent test for the quality (e.g. proper folding) of each of the CR domains. Can the authors show whether these bind calcium and indeed fold properly? This would be important to concluding that only CR2 and CR3 are active in G interactions.

- The authors determine two crystal structures of the CR2/3:G complexes, but there seems to be minimal analysis on these. For example, the authors should indicate the buried surface area in each complex and provide some overall analysis of the interfaces, including number of residues involved, contacts made with each CR domain and the relative conservation of contacts (this is partly shown in Figure 4, but not explained in the text). The interface appears small and not particularly well conserved.

- Based on the crystal structures and sequences of the other LDL-R CR domains, can the authors explain the specificity of the observed interactions with CR2/3? How about the potential interactions with other CR domains, for example those in other LDL-R family members? There was no effort to examine specificity of the interactions across these many potential interacting CR domains, which seems particularly of interest given the fact that VSV can utilize different receptors for entry.

- How many CR domains in other proteins might be able to interact with G? The authors suggest that their data indicate that other LDL receptor family members may act as receptors, but the experimental test of this possibility is limited (RAP inhibition). Furthermore, RAP inhibition is incomplete suggesting that it does not block all receptor interactions. Could G interact with other CR domain containing proteins outside of this family?

- The authors study a number of G mutants at the CR binding interface. The K47A mutant shows a select defect in CR3 binding. Is there a structural explanation for this result? In general, the authors should better correlate the functional data with the structures to explain the observed CR binding defects.

- The authors only check the fusion activity of two of the mutants made (K47A and R354A), but the other mutants should also be tested for activity in the fusion assay to preclude their possible

destabilization and folding to the postfusion form. Furthermore, it is unclear in this assay what percentage of G would need to be active to give "wt" results. It would be useful if the authors could independently check the proper folding (e.g. pre-fusion rather than post-fusion) of each mutant using (for example) conformation specific antibodies.

- On page 11 last paragraph, the authors state that "The demonstration that the receptors of VSV are all members of the LDL-R family..." I do not think the authors have demonstrated this. As indicated above the RAP inhibition used to look at LDL-R family members show incomplete inhibition. It is unclear whether other CR domains in unrelated proteins could interact with G and there is no specificity analysis of the CR domain interaction that supports the conclusion that G only interacts with CR domains within the LDL-R family. The authors have shown that residues in G that mediate CR domain interactions are critical for interactions with VSV receptors, but that does not directly address the issue of LDL-R family specificity

Additional minor points.

Page 6, first paragraph. The authors report K_d values in the text that are not the same as the ones in the figure. Although the discrepancy is explained in the figure legend (because of replicates), this should be incorporated (briefly) into the main text.

Page 6, last paragraph. "Those three segments are scattered..." Probably rearranged or reorganized would be clearer than scattered in this context.

Reviewers' comments:

Reviewer #1

The current study investigates the structural basis for recognition of VSV glycoprotein by LDL receptor family members. This is an important and significant study for two reasons: First, the study adds to our knowledge of ligand recognition by this family of receptors, and second, provides structural insight into how vesicular stomatitis virus interacts with host cell receptors. Overall, the study appears to be carefully done. However, attention to some minor details would improve the study significantly.

1. The CR repeats from the LDL receptor were prepared in bacteria as a fusion protein with GST. Were these fragments subjected to a refolding process, which would be necessary to obtain correctly folded domains? This is particularly important for the data generated from Figure 2.

The GST-CR proteins were indeed subjected to a refolding process, adapted from Harper and Speicher (2011), which was already described in the methods section in the paragraph dedicated to protein expression purification and labeling. Sarkosyl, which plays the role of mild denaturing agent, and DTT in the lysis buffer as well as Ca^{2+} all along the purification are mandatory for correct folding. We have now added supplementary data (Supplementary Fig. 2) showing that all the purified CR domains behave similarly in gel filtration experiments (See also our answer to the first comment of reviewer #3).

2. The binding of VSV glycoprotein G to CR2 and CR3 is understandably weak (Kd values in the μ M range). However, the binding of VSV to cellular receptors presumably occurs with much higher affinity. How does this occur? Is it a result of multivalent interactions? The authors should discuss this and perhaps revise the model depicted in Fig 3e to describe this more precisely.

The data of Finkelshtein et al. (PNAS, 2013) indeed suggested a much higher affinity of LDLR for G. Here, we have only analyzed the binding of isolated CR domains with G. It is plausible that the affinity increases when CR domains are presented in the context of the full receptor. In the context of the virus particle, an avidity phenomenon is also expected as several glycoproteins can bind several receptors. However, the binding of two G protomers to a single LDL-R is excluded as CR2 and CR3 are in too close proximity. The cartoon depicted in Fig. 3E is here to show that the CR domains have the correct orientation to interact with G in the viral membrane (and nothing else).

3. The authors conclude from the experiments depicted in Figure 6 that other LDL receptor family members participate as receptors for VSV. It would be of interest to determine which LDL receptor family members are expressed in these HAP-1 cells. This could be done by PCR for example.

We have made a systematic search of CR domains having characteristics similar to CR2 and CR3 (see also our answer to the fourth comment of reviewer #3). We found such CR domains in human VLDLR, LRP1, LRP1B, LRP2, LRP3 and LRP4 (new Fig. 7E). Using RT-PCR, we

have confirmed that those receptors (except LRP2) are indeed expressed in HAP-1 LDL-R^{KO} cells (new Fig. 6C).

4. In the experiments shown in Figure 6B, the RAP concentration should be given.

This is now given in the legend of the figure.

Reviewer #2

Albertini and colleagues report the crystal structures of VSV G in complex with two cysteine-rich domains (CR2 and CR3) of the LDL receptor. Both CR2 and CR3 bind to the same site on VSV G. Mutational analyses of the G binding site shows that VSV loses its infectivity, thus indicating that LDL-R constitutes an essential receptor for VSV entry. The main results are:

- CR2 and CR3 binding to G is pH dependent.
- CR2 and CR3 bind G in the low micromolar range.
- CR2 and CR3 domains block VSV G infection in a cellular model.
- The co-crystal structures define the interaction interface with contributions from TRD, PHD and S2. These regions do not form a continuous surface in low pH G.
- Two basic residues of G have been identified as key residues for interaction.
- Knock-down of LDL-R indicates that HAP-1 cells can be still infected with VSV. The authors conclude that alternative receptors may allow infection. However, the western blot of fig. 6 shows a faint band in the knock-down cells and this residual LDL-R expression may be sufficient for infection. In addition, RAP only blocked VSV infection to 50% in the knockdown cells. This should be considered in the discussion of alternative receptors. G proteins carrying mutations of the basic residues K47 and R354 do not rescue VSV infection, corroborating the importance of an intact receptor interaction surface.

After having received this review, we have been informed by the purchaser (Horizon) that there might be some problems with some LDL-R invalidated HAP-1 cell clones they provided. Therefore, besides HAP-1 LDL-R^{KO2} that we previously used, we have included a second HAP-1 LDL-R^{KO} (HAP-1 LDL-R^{KO1}) cell line. For both clones, we sequenced the LDL-R gene invalidated regions (new Supplementary Fig. 4). We observed a deletion of 2nt in the coding region of HAP-1 LDL-R^{KO2} and a deletion of 8 nt in the coding region of (HAP-1 LDL-R^{KO1}). Therefore, in both cell lines, the LDL-R gene is indeed invalidated in agreement with the data presented in a new western blot in which no LDR-R expression was detected (Figure 6A). There is indeed a faint and diffuse band in the lane corresponding to the sample from HAP-1 LDL-R^{KO2}. This band migrates slightly below the one corresponding to wild type and mature LDL-R and might be the result of a rare alternative splicing of mRNA.

The experiments presented in figure 6B were therefore performed again and indeed confirm the fact that RAP only blocked ~50% of VSV infection in KO cells. This is not surprising considering (i) that RAP was used only at a concentration of 50nM (because of limited amount of protein available) and (ii) that RAP does not bind all CR domains with the same

affinity. As a consequence, in the experimental conditions used, RAP might not block all the CR domains available at the cell surface.

In summary, the manuscript is well written (some editing required) and of high technical quality. The results are presented and discussed in a clear way and they support the major conclusions. The work thus represents an important step forward in understanding the molecular details of VSV G interaction with its cellular receptors.

Minor comments:

The R.m.s. deviation of the bond angles (1.8°) is rather high in the 2.25 Å structure! The line indicating the resolution range of the processed data needs to be shifted in the table.

We ran additional refinement cycles. Now the R.m.s. of the bond angles is 0.987° .

Reviewer #3

This manuscript reports structural and functional studies of the VSV G protein with the LDL receptor. The LDL receptor is a receptor for VSV entry and interacts directly with the VSV G protein. Here the authors dissect the interactions of the LDL receptor with VSV G, by expressing individual CR domains. Only CR2 and CR3 are shown to interact with prefusion G and crystal structures of both complexes are obtained. Based on the crystal structure analysis, residues of G which interact with the calcium binding site on the CR domains are implicated in receptor binding and entry. Mutagenesis of G demonstrates that these interactions are critical for LDL-R binding as well as more generally for proper G function in entry. These data indicate that other receptors that mediate VSV entry form similar interactions and implicate other LDL-R family members as potential receptors. Although overall the manuscript is well written and the data is convincing, there are a number of issues with the current version that need to be addressed.

- Individual CR domains from the LDL-R are expressed for analysis and the data indicate that only CR2 and CR3 interact with G. However, there is no independent test for the quality (e.g. proper folding) of each of the CR domains. Can the authors show whether these bind calcium and indeed fold properly? This would be important to concluding that only CR2 and CR3 are active in G interactions.

As mentioned in our answer to the first comment of reviewer 1, the GST-CR proteins were subjected to a refolding process which was described in the Methods section in the paragraph dedicated to protein expression purification and labeling. The same refolding protocol was used for each CR domain.

In absence of calcium, GST-CR2 and GST-CR3 do not fold properly and, as a consequence, do not bind VSV G.

We have now added supplementary data (Supplementary Fig. 2) showing that all the refolded CR domains behave similarly in gel filtration experiments on a S75 Superdex HR10/30

column. First, the elution profile of the refolded CR domains reveals a single peak at 14.25 +/- 0.25 ml. When previously denatured in presence of EDTA and DTT, all the CR domains eluted about 0.75 ml earlier (Supplementary Fig. 2).

- The authors determine two crystal structures of the CR2/3:G complexes, but there seems to be minimal analysis on these. For example, the authors should indicate the buried surface area in each complex and provide some overall analysis of the interfaces, including number of residues involved, contacts made with each CR domain and the relative conservation of contacts (this is partly shown in Figure 4, but not explained in the text). The interface appears small and not particularly well conserved.

We have now indicated the buried surface area (~1500Å²) in the text. The interface is indeed small. This is consistent with the dissociation constant in the 1-10 μM range.

The contacts were indeed described in detail in Figure 4. The important features were also presented in the last paragraph of the section entitled “Crystal structures of G_{th} in complex with LDL-R CR domains”. Nevertheless, we have added a sentence to mention the interaction between G residue K47 and Q71 (resp D110) in CR2 (resp. CR3) which was only visible in figure 4 and not described in the text.

- Based on the crystal structures and sequences of the other LDL-R CR domains, can the authors explain the specificity of the observed interactions with CR2/3? How about the potential interactions with other CR domains, for example those in other LDL-R family members? There was no effort to examine specificity of the interactions across these many potential interacting CR domains, which seems particularly of interest given the fact that VSV can utilize different receptors for entry.

The specificity of the interaction is not easy to explain. All CR domains have the same fold and all form a calcium cage involving conserved acidic residues. The acidic residues in position I and II, which play a key role in the interaction with G, are conserved (except in CR7 which has an asparagine in position I). Therefore, they cannot be discriminant. Furthermore, residue R354 of G essentially interacts with the main chain of both CR2 and CR3. Here again, the influence of CR domains amino acid sequence on those interactions is difficult to predict.

However, we note that the aromatic residues (W66 in CR2 and F105 in CR3) establishing hydrophobic interactions with the aliphatic part of K47 side chain and with the side chain of A51 of G are not always conserved and are replaced by an arginine in CR6 and a lysine in CR7. Also, instead of an amide or an acidic amino acid in place corresponding to residue Q71 in CR2 and D110 in CR3, a serine is found in CR1, a glycine in CR5 and an alanine in CR7. Those considerations may explain why we do not observe G binding to CR1, CR5, CR6 or CR7. The question remains opened for CR4. This is now presented in the discussion.

- How many CR domains in other proteins might be able to interact with G? The authors suggest that their data indicate that other LDL receptor family members may act as receptors, but the experimental test of this possibility is limited (RAP inhibition). Furthermore, RAP

inhibition is incomplete suggesting that it does not block all receptor interactions Could G interact with other CR domain containing proteins outside of this family?

We have now made a systematic search of CR domains having similar characteristics to CR2 and CR3 (i.e. besides the common characteristics of the CR domains, they must have no insertion or deletion in the contact region, an aromatic residue corresponding to W66 of CR2 to interact with the aliphatic part of K47, and an amide or an acidic amino acid in place corresponding to residue Q71 of CR2). In humans, we found two such CR domains in VLDLR, two in LRP1, one in LRP1B, three in LRP2, one in LRP3, one in LRP4, one in PGBM (Basement membrane-specific heparan sulfate proteoglycan core protein). In Drosophila, we found a single CR domain with those characteristics in the putative vitellogenin receptor. This is now presented in new Figure 7E and discussed in the text.

Although other proteins from other families (such as the corin, which is a transmembrane serine protease) also contain CR domains, their CR domains do not seem to have the correct characteristics to interact with G (and are not retrieved in our systematic search). Furthermore, Corin is a type II membrane protein imposing the opposite orientation of CR domains, a feature which may also impede G binding.

Concerning the data on RAP inhibition, see our answer to reviewer #2.

- The authors study a number of G mutants at the CR binding interface. The K47A mutant shows a select defect in CR3 binding. Is there a structural explanation for this result? In general, the authors should better correlate the functional data with the structures to explain the observed CR binding defects.

Although some binding is still observed, K47A mutant clearly binds less efficiently CR2 (Figure 5A). It is very difficult to propose a structural explanation for such small differences.

We have described the importance of key residues in the section entitled “Crystal structures of G_{th} in complex with LDL-R CR domains”. The following section entitled “K47 and R354 of G are crucial for LDL-R CR domains binding”, in which we mutate the key residues and describe the associated phenotypes, is following quite logically. We think that, in this case, the correlation between functional data and structures is obvious. However, we have expanded the discussion section on this topic.

- The authors only check the fusion activity of two of the mutants made (K47A and R354A), but the other mutants should also be tested for activity in the fusion assay to preclude their possible destabilization and folding to the postfusion form. Furthermore, it is unclear in this assay what percentage of G would need to be active to give “wt” results. It would be useful if the authors could independently check the proper folding (e.g. prefusion rather than postfusion) of each mutant using (for example) conformation specific antibodies.

We have now presented the fusion activity of all the mutants (Figure 5D). They all have kept their fusion activity, indicating that they are correctly folded. The debate on the percentage of

G which is required to give wild-type results in a specific fusion assay is out of the scope of this work.

Of note, we have now quantified the incorporation of G into pseudotypes (in the previous version, we only presented a western blot). This confirms that mutant Gs were incorporated as efficiently as wt G in the viral particles (Figure 7B and 7C).

- On page 11 last paragraph, the authors state that “The demonstration that the receptors of VSV are all members of the LDL-R family...” I do not think the authors have demonstrated this. As indicated above the RAP inhibition used to look at LDL-R family members show incomplete inhibition. It is unclear whether other CR domains in unrelated proteins could interact with G and there is no specificity analysis of the CR domain interaction that supports the conclusion that G only interacts with CR domains within the LDL-R family. The authors have shown that residues in G that mediate CR domain interactions are critical for interactions with VSV receptors, but that does not directly address the issue of LDL-R family specificity

We carefully stated that this is “the most parsimonious interpretation” of our data. We have now added a paragraph about other transmembrane proteins which contain CR domains but are not members of the LDL-R family.

Of course, we cannot formally exclude that the G residues involved in CR domains recognition are also involved in the recognition of a completely unrelated receptor.

Additional minor points.

Page 6, first paragraph. The authors report Kd values in the text that are not the same as the ones in the figure. Although the discrepancy is explained in the figure legend (because of replicates), this should be incorporated (briefly) into the main text.

Done.

Page 6, last paragraph. “Those three segments are scattered...” Probably rearranged or reorganized would be clearer than scattered in this context

Done.

REVIEWERS' COMMENTS:

Reviewer #1 (Remarks to the Author):

The authors have addressed all of my concerns in the revised manuscript. Overall, this is a well done and important study that gives considerable structural insight into ligand recognition by LDL receptor family members and furthers our understanding of how vesicular stomatitis virus interacts with host cell receptors.

Reviewer #3 (Remarks to the Author):

The authors have addressed all of my comments in the revised manuscript.